# Gender-based homophily in collaborations across a heterogeneous scholarly landscape

Y. Samuel Wang[1�]*, Carole J. Lee[2], Jevin D. West[3], Carl T. Bergstrom[4], Elena A. Erosheva[5☯]

1 Department of Statistics and Data Science, Cornell University, Ithaca, NY, United States of America, 2 Department of Philosophy, University of Washington, Seattle, WA, United States of America, 3 Information School, University of Washington, Seattle, WA, United States of America, 4 Department of Biology, University of Washington, Seattle, WA, United States of America, 5 Department of Statistics, University of Washington, Seattle, WA, United States of America

☯ These authors contributed equally to this work.

* ysw7@cornell.edu

**Data Availability Statement:** The data is owned by JSTOR, and JSTOR must grant permission for usage. See https://www.jstor.org/contact-us/ for contact details.

## Abstract

In this article, we investigate the role of gender in collaboration patterns by analyzing gender-based homophily—the tendency for researchers to co-author with individuals of the same gender. We develop and apply novel methodology to the corpus of JSTOR articles, a broad scholarly landscape, which we analyze at various levels of granularity. Most notably, for a precise analysis of gender homophily, we develop methodology which explicitly accounts for the fact that the data comprises heterogeneous intellectual communities and that not all authorships are exchangeable. In particular, we distinguish three phenomena which may affect the distribution of observed gender homophily in collaborations: a *structural* component that is due to demographics and non-gendered authorship norms of a scholarly community, a *compositional* component which is driven by varying gender representation across sub-disciplines and time, and a *behavioral* component which we define as the remainder of observed gender homophily after its structural and compositional components have been taken into account. Using minimal modeling assumptions, the methodology we develop allows us to test for behavioral homophily. We find that statistically significant behavioral homophily can be detected across the JSTOR corpus and show that this finding is robust to missing gender indicators in our data. In a secondary analysis, we show that the proportion of women representation in a field is positively associated with the probability of finding statistically significant behavioral homophily.

## 1 Introduction

Scholarly collaboration now drives the frontier of knowledge as well as the complexity, hyper-specialization, and funding of research [1–9]. Given the professional advantages of collaboration, it is not surprising that the relative rate of collaborative research has increased over time and now dominates solo-authorships in all areas except for the humanities [10], with rates of collaboration rising even there [11]. As in other professional relationships, researchers choose

**Funding:** This research was supported by the Royalty Research Fund Grant #A118374 awarded to EE (PI) and CL (co-PI), National Science Foundation Grant #1735194 awarded to JW (co-PI), and National Science Foundation SMA 19-52069 to CTB. https://www.washington.edu/research/or/royalty-research-fund-rrf/ https://www.nsf.gov/ The funders had no role in study design, data collection and analysis, decision to publish, or preparation of the manuscript.

**Competing interests:** The authors have declared that no competing interests exist.

collaborators based on instrumental considerations such as expertise and resources [6–8], as well as social features such as respect, trust, personal chemistry, and friendship [7–9]. Homophily—the principle that similarity breeds connection between individuals—has been found to structure professional as well as socio-emotional relationships [12]. Gender homophily in particular creates profound divides in work environments [13], voluntary associations [14], and friendships [15, 16].

Existing work reveals gender gaps in collaborative teams. Women are less likely than men to co-author [11, 17], and when they do, they are less likely to occupy the first or last author positions—positions of power and prestige [18, 19]. When women miss opportunities to participate and take leading roles in collaborations, they miss opportunities to develop mentoring relationships, knowledge, skills, and professional credentials relevant for grant-getting, hiring, promotion, and tenure [1–4].

Previous research on gender homophily in scholarly team formation has been limited by reliance on existing academic structures. Early studies considered individual academic disciplines such as economics: a study of 178 PhDs in economics found that women were over five times more likely than men to have women co-authors [20]; another study of 3,090 articles in the top economics journals from 1991–2002 found evidence of gender homophily at the subfield level [21]. More recently, gender homophily has been found across the life sciences [22] and beyond [23]; however, even these latest studies only account for a coarse and discipline-driven structure of the scholarly landscape.

The primary goal of our analysis is to test the hypothesis that the gender homophily observed in the existing scholarly corpus can be fully explained by the varying gender representation across intellectual communities and time. In particular, one would reasonably expect individuals to collaborate with individuals who hold similar intellectual interests, and empirically, we see that gender representation varies widely across time and across different disciplines and even across sub-communities within a single discipline. Thus, even if individuals are selecting co-authors without regard to gender, the propensity to collaborate with other researchers with similar interests will result in observed gender homophily in co-authorships. This phenomenon has been noted in other settings. For instance, in population genetics, this phenomenon would be an example of the Wahlund effect [24] where separate sub-populations may each be matching non-assortatively, but the population—when considered as a whole—deviates from what would be expected under a Hardy-Weinberg equilibrium because there is no mixing between sub-populations. We do not aim to otherwise draw affirmative causal conclusions about why gender homophily in scholarly collaborations may occur in these communities; however, a secondary analysis provides interesting associative results on the characteristics of fields.

In this work, we consider a broad scholarly landscape—the JSTOR corpus, an archive that includes documents across the physical sciences, social sciences, and humanities. While previous studies have also considered a broad corpus of documents [23], the novelty of our work lies in a method which allow us to exploit a previously developed hierarchical clustering of science [25] which partitions the scholarly landscape at varying levels of data-driven granularity—from two dozen discipline-like fields to over 1,400 small intellectual communities. Given the characteristics of each small intellectual community and how it is positioned in the hierarchical clustering, the procedure we develop accounts for the observed homophily which could be attributed to variations in gender representation and authorship norms across sub-disciplines and scholarly fields. Thus, we explicitly test for the presence of gender homophily beyond what might occur due to heterogeneity of gender representation across sub-disciplines and time. Put succinctly, our findings show that, when analyzing a large corpus of documents

and respecting the heterogeneity of the underlying structure, the observed gender homophily cannot be fully explained by differing gender representation across sub-disciplines and time.

## 2 Problem setup

### 2.1 Quantifying homophily

We consider the unit of analysis to be an *authorship*—an instance of co-authoring a single document—rather than an *author*—who may have co-authored multiple documents. The implications of this decision are addressed in Section 6, and we recognize that using authors could yield a higher fidelity model. However, given that "fully disambiguated authors" is difficult to achieve, we believe a more conservative approach is to use authorships and clearly state the shortcomings, rather than use poorly disambiguated authors and overstate results.

For a *corpus*—a set of documents and authorships—we can calculate the *index of assortativity* [26], denoted by $\alpha$, which is a measure of homophily between [−1, 1] where larger (more positive) values indicate more homophily and smaller (more negative) values indicate heterophily. Specifically, the index of assortativity is defined to be $\alpha = p - q$, where $p$ is the probability that a randomly selected co-authorship of a randomly selected man authorship is a man, and $q$ is the probability that a randomly selected co-authorship of a randomly selected woman authorship is a man. Setting the reference gender as women—i.e., redefining $p$ to be the probability that a randomly selected co-authorship of a randomly selected woman authorship is woman and redefining $q$ to be the probability that a randomly selected co-authorship of a randomly selected man authorship is woman—yields an equivalent value. If a set of authorships only contains authorships of one gender, $\alpha$ is undefined. Our analysis framework is not dependent on a particular metric; however, we use $\alpha$ due to its interpretation as a difference in risks and its connection to Wright's coefficient of inbreeding [27]. Indeed, $\alpha$ is a generalization of Wright's coefficient to the multi-author scenario, and the two measures are equivalent when all documents have two authors. Furthermore, $\alpha$ is equal to the observed coauthor-gender correlation in a given collection of documents [26], and $\alpha$ is equivalent to Newman's network-based assortativity coefficient [28] in an appropriately weighted network [29].

For concreteness, consider a corpus of documents where all documents have two authors, and let $0 \leq \pi \leq 1/2$ be the proportion of the less frequent gender. If the proportion of woman-man documents is $2\pi(1 - \pi)$—roughly what we would expect under random pairings—then $\alpha = 0$; if there are no woman-man documents, $\alpha = 1$; if the proportion of woman-man documents is $2\pi$—the largest attainable value—then $\alpha = -\pi/(1 - \pi)$ which is between −1 and 0. Details on these calculations are given in the supplement.

To measure practical impact and allow for a comparison of different $\alpha$ values, we can also convert $\alpha$ back into a more interpretable quantity. Consider a setting where every field consists of 100 two-author documents. Then for a given $\alpha$ and proportion of woman authorships $\pi$, we can calculate the number of heterophilous (woman-man) documents: $WM(\alpha, \pi) = 200(1 - \alpha)\pi(1 - \pi)$. This gives concrete way to contextualize and compare two different values of $\alpha$.

### 2.2 Homophily in a heterogeneous setting

The metrics we defined in the previous section only measure homophily through outcomes, but do not reflect the intent or motivations of the individual authors. Thus, while we can measure whether or not individual authorships co-authored with others of the same gender, we cannot directly measure whether they did so directly due to gender or if the outcome was the result of other factors. We use *observed gender homophily* to indicate the observed value of $\alpha$—regardless of the process (or intent of authors) under which co-authorships were formed.

In a corpus of documents, if authorships (implicitly or explicitly) prefer co-authors of the same gender, we would expect to observe a large positive value of $\alpha$. However, a large value of $\alpha$ may be observed due to variety of reasons, so determining a cut-off when $\alpha$ is "large enough" to suggest that gender directly affects co-authorships is not straightforward. Thus, we use a statistical hypothesis testing framework to test the null hypothesis that the observed gender homophily can be fully explained by varying gender composition across intellectual communities and time. In this section, we first build intuition and motivate the need for novel methodology; the details of our testing framework are given in Section 4.

To test the null hypothesis that the observed gender homophily can be fully explained by heterogeneity across sub-disciplines and time, we compare the observed value of $\alpha$ to values drawn from a *null distribution*. The null distribution generates plausible alternative values of $\alpha$ which might have been observed under the null hypothesis. If the observed $\alpha$ is larger than most of the values of $\alpha$ from the null, then the observed value would be "unlikely to occur" under the null distribution, and we might conclude that the null hypothesis is false. This would imply that other factors—potentially including authorship gender—affected co-authorship formation. However, a realistic distribution of plausible alternative $\alpha$ values—even if gender does not play a direct role in co-authorship formation—depends on many aspects of the corpus of documents it is measured on. To elucidate this point, we distinguish between the *structural*, *compositional*, and *behavioral* aspects which may affect the distribution of $\alpha$. Throughout this discussion, we will use the term *field* colloquially to refer to a corpus of authorships focused on a topic and *sub-field* to refer to a subset of authorships focused on an even narrower topic.

Consider a corpus which comprises a narrow sub-field or tightly focused intellectual community. Specifically, suppose the topic of study is sufficiently narrow so that each observed authorship could have co-authored with any other authorship in the sub-field. To generate plausible alternative configurations of the authorships and documents where gender does not directly affect co-authorship decisions, we could randomly permute the authorships so that each one is assigned to a (potentially) different document with (potentially) different co-authors. Under this process the expected value of $\alpha$ should be near (though not exactly) 0, and the expected value will tend to 0 as the number of authorships in the corpus grows. However, the shape of the distribution can still vary quite a bit depending on the structural aspects such as gender distribution, the number of documents and authorships, and the number of authorships for each document. The *structural homophily distribution* for a corpus of documents (either field or sub-field) is the distribution of $\alpha$ which results from randomly permuting all authorships within the corpus.

Fig 1 provides an illustrative example of two sub-fields, *A* and *B*. In Fig 2, the top two panels show the distributions of $\alpha$ for sub-fields *A* and *B* when authorships are randomly permuted to documents within their own sub-field. The expected values of $\alpha$ for sub-fields A and B are not 0, but −0.08 and −0.13 respectively. Furthermore, we see that the distribution of $\alpha$ is quite different in sub-field *A* when compared to sub-field *B*. For sub-field *B*, every possible permutation of the authorships results in the same value of $\alpha$, while sub-field *A* exhibits a range of possible values.

When considering a field which aggregates multiple sub-fields, the assumption that authorships are uniformly likely to co-author with any other authorship in the corpus is no longer reasonable because individuals will tend to co-author with others who are in the same sub-field and share intellectual interests. Thus, configurations of the corpus formed by randomly permuting all authorships may no longer be plausible alternatives, and for any pair of authorships, the probability of co-authorship ought to depend on the sub-field in which they were originally observed. In addition, we observe empirically that gender composition differ widely

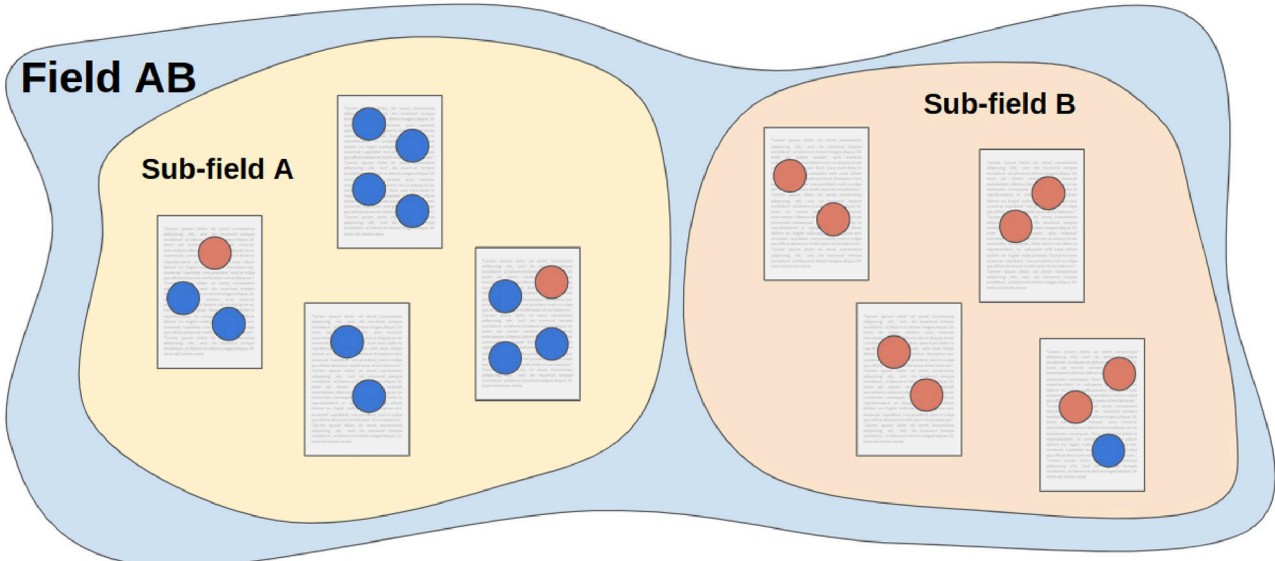

**Fig 1. An example corpus.** Each square represents a document, and each circle represents an authorship with the color indicating gender. Sub-fields A and B both exhibit negative $\alpha$, −2/11 and −1/8 respectively. However, when all documents are aggregated together, Field AB has a positive $\alpha = 9/20$. Compositional homophily drives this counterintuitive result.

across sub-fields—even sub-fields within the same field. Thus, the gender of each authorship is also associated with sub-field membership.

This causes an additional phenomenon which can result in large observed values of $\alpha$ even if teams are formed without regard to authorship gender. Specifically, when authorships within the same sub-field are more likely to be the same gender, homophily with respect to sub-field may induce observed gender homophily in an aggregated corpus. As a result, when $\alpha$ is calculated on a field which aggregates sub-fields with varying gender representation, we would expect a larger observed value of $\alpha$ even if gender does not play a direct role in co-authorship choices. *Compositional homophily* is the phenomenon by which homophily with respect to sub-field causes observed gender homophily in co-authorships (i.e., results in a larger value of $\alpha$) because of varying gender distributions across sub-fields. Said in probabilistic terms, although the gender of co-authors may be conditionally independent given their sub-field, they will generally be dependent when we do not condition on sub-field membership (i.e., do not account for sub-field membership). Indeed, in Fig 1, the observed values of $\alpha$ are −2/11 and −1/8 in sub-field A and B respectively; however, when considering field AB and all 8 documents are aggregated, $\alpha = 9/20$.

We emphasize that under the null hypothesis, the distribution of $\alpha$ will reflect both the structure of the corpus and the effect of compositional homophily. However, in contrast to structural homophily, when compositional effects are present, the expected value of $\alpha$ when permuting authorships does not tend to 0 as the number of authorships in a field increases. In Fig 2, we show two distributions of $\alpha$ measured over field AB. In the bottom left panel, we only account for structural homophily and uniformly permute all authorships to any document in the field (regardless of the original sub-field). The shape of this distribution depends on the structural aspects of the field, but the expected value of $\alpha$ under this distribution is close to 0. Notably, we see that when the null distribution only reflects the structural homophily of the corpus, the observed value of $\alpha$ for field AB is unlikely to occur and one might conclude that other factors may have affected the process which generated the observed $\alpha$.

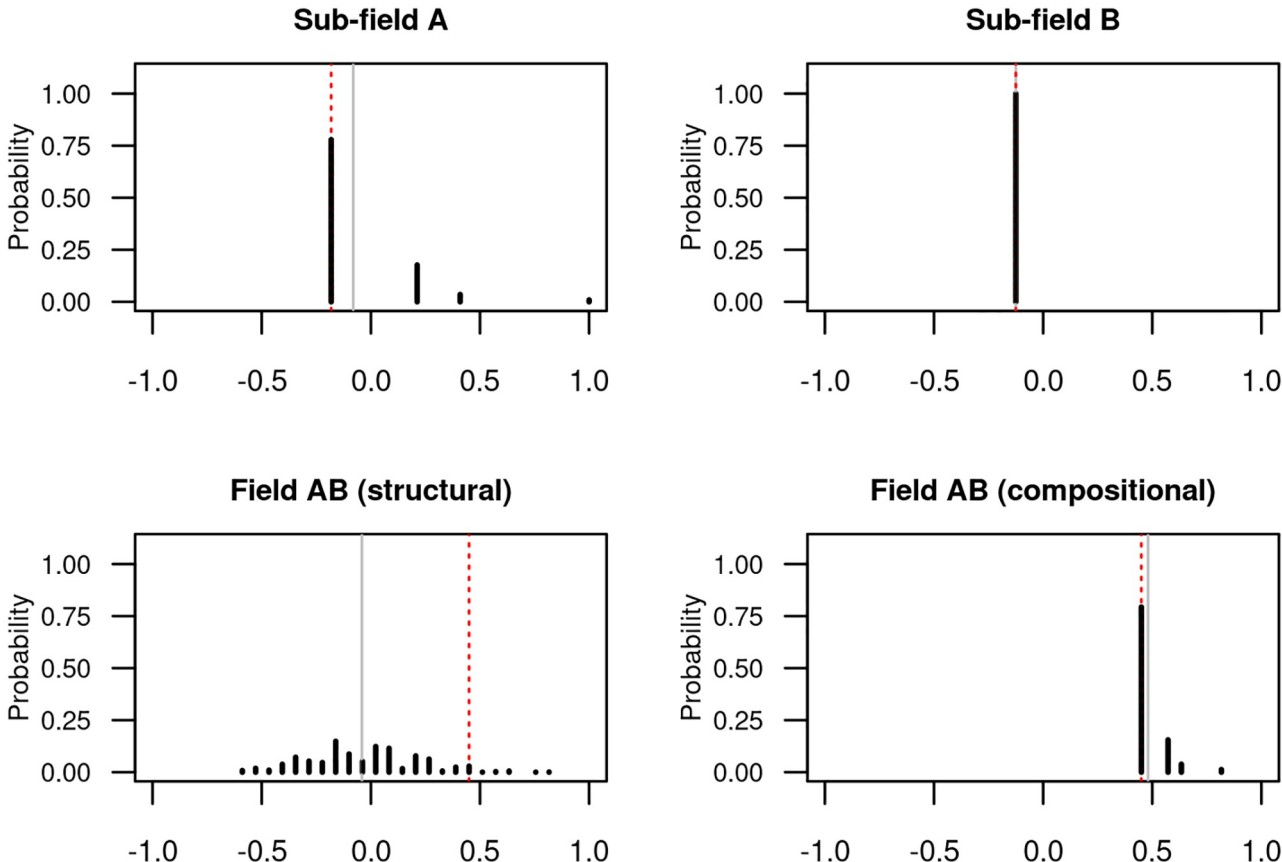

**Fig 2. Structural and compositional homophily example.** The plot shows the distribution for the index of assortativity, $\alpha$ when the authorships are generated according to several processes. The top panels show the distribution of $\alpha$ which reflects the structural homophily for sub-fields A and B; i.e., authorships are randomly permuted to different documents. The bottom left panel shows the distribution of $\alpha$ for the field AB which only reflects structural homophily (i.e., all authorships in the field are randomly permuted) and the bottom right panel shows the distribution of the field which reflects both structural and compositional homophily (i.e., authorships are randomly permuted within their sub-field). In each plot, the dotted red line indicates the observed value of $\alpha$ and the solid gray line indicates the mean value under the null distribution. Note that under the null distribution which only reflects structural homophily, the observed $\alpha$ is larger than the mean $\alpha$. However, in the null distribution which accounts for compositional homophily, the mean $\alpha$ is larger than the observed $\alpha$.

However, suppose that the topics studied in sub-fields A and B are sufficiently distinct so that co-authorships between sub-fields are unlikely to occur. Due to compositional homophily, because the gender distributions in sub-fields A and B are quite different, we would expect larger values of $\alpha$ to occur even under the null hypothesis. The bottom right panel shows the distribution of $\alpha$ which accounts for this. Specifically, we respect the sub-field structure by randomly permuting each authorship to a document *within its original sub-field*, but measure $\alpha$ on the entire aggregated field. The expected value of $\alpha$ under this distribution is no longer close to 0. Indeed, we see that the observed value of $\alpha$ = 9/20 for the aggregated corpus is no longer unusual and may be plausibly explained by sub-field homophily alone (i.e., the outcome would not be unusual under the modeled compositional homophily).

Put succinctly, structural homophily reflects the distribution of $\alpha$ we would expect when all co-authorships are selected from a pool with equal probability. However, this "baseline" depends on various attributes and is different for each specific intellectual community. Compositional homophily reflects the distribution of $\alpha$ when co-authorship choices do not depend on gender, but depend on attributes—such as sub-field membership or intellectual interests—

which may be associated with gender. Blau [30] uses "consolidation" to describe homophily induced by factors (in this case sub-field membership) which are associated with the factor of interest (in this case gender). However, this general phenomena also appears in other fields. In statistics, it is a case of Simpson's paradox [31] where homophily is confounded by gender imbalances across scholarly fields. In population genetics, this phenomenon is known as the Wahlund effect [24]: random mating within subpopulations does not imply Hardy-Weinberg equilibrium in the population as a whole.

The primary methodological innovation of our work consists of accounting for compositional homophily due to sub-field structure. However, women representation has increased over time in almost all disciplines (see plots of women authorships over time in the supplement) and co-authorships can only occur if researchers are active at the same time. Thus, just as aggregating sub-fields with differing gender representation can induce compositional homophily, considering a corpus of documents across a range of years with differing gender representation can also induce compositional homophily. Thus, as we will discuss in Section 4, we also explicitly account for publication years when generating plausible alternatives.

In contrast to structural and compositional homophily which could occur when co-authors are selected irrespective of gender, we use *behavioral homophily* to describe deviations of $\alpha$ from its distribution under structural and compositional homophily. This notably may include the effect of both explicit and implicit consideration of gender when selecting co-authors. However, this may also be due to consideration of other factors including race, ethnicity, or career stage. We further discuss these alternative factors in Section 6. We use the term "behavioral" to indicate that it is divorced from any claims about author intent, but a reflection of observed behavior.

These notions of homophily map onto the two components of homophily discussed by McPherson et. al. [12]—baseline homophily and inbreeding homophily—in a context of voluntary and professional network ties such as those of friendship, support, and advice. Specifically, they described baseline homophily as homophily "created by the demography of the potential tie pool" and inbreeding homophily as "homophily measured as explicitly over and above the opportunity set" [12]. Our notion of *structural homophily* aligns with baseline homophily of McPherson et. al. [12], though we prefer to use the term "structural" which does not have temporal connotations (i.e., baseline observations in longitudinal studies). McPherson et. al. [12] emphasize that their definition of inbreeding homophily does not refer to "choice homophily purified of structural factors," but instead encompasses "homophily induced by social structures below the population level . . . to homophily induced by other dimensions with which the focal dimension is correlated, and to homophily induced by personal preferences." [12]. Indeed, *compositional homophily* accounts for homophily induced by "structures below the population level" and the correlated dimension of intellectual interests, and *behavioral homophily* may include aspects "induced by personal preferences."

## 3 Data

The JSTOR corpus includes documents from over 2,500 journals spanning social science, natural sciences, and the humanities. The raw dataset we start with includes 1,450,605 papers with 2,787,833 authorships published from 1665–2011. To identify the sub-structures in the JSTOR corpus which may induce compositional homophily, we apply a hierarchical implementation of the InfoMap network clustering algorithm to the citation network on the corpus [25, 32]. The algorithm reveals the hierarchical structure of the corpus through efficient coding of random walks on the citation network. At the lowest level of the clustering, each document is grouped into one of 1,450 *terminal fields* which form the finest partition of the data. These

terminal fields are indicative of scholarly communities tied by shared narrow research topics or methodologies. In the time period we consider (1960–2011), the median number of total documents in each terminal field is 102 (roughly 2 documents a year), and 95% of terminal fields have fewer than 570 total documents (roughly 11 documents each year). Each higher level of the clustering forms a progressively coarser partition of the documents by aggregating terminal fields into *composite fields*. The hierarchical structure obtained from the InfoMap algorithm has up to 6 levels. At the top level of the clustering, there are 24 identified *top-level fields* that are indicative of disciplinary divisions such as molecular and cell biology, economics, statistics, and sociology. We specifically choose to analyze JSTOR (as opposed to other possible databases) because of the previously developed heirarchical clustering which has been painstakingly hand labeled [19]. This allows for interpretable results which may also be useful for generating hypotheses for future work. We will use "field" to refer to a corpus which may be a top-level, composite, or terminal field. At any given level of hierarchy, documents in a common field are more connected to each other via citations than they are to documents from neighboring fields. Likewise, the fields defined by a lower (finer) level of the hierarchy are more connected than fields defined by a higher (coarser) level in the hierarchical clustering. This hierarchical clustering allows us to test for behavioral homophily at varying levels of granularity and shows how a larger field might be composed of several smaller sub-fields. In particular, we can increase the power of our statistical tests by considering larger portions of the corpus while still accounting for the confounding due to compositional homophily induced by hetereogeneity of the larger set of documents. An interactive browser of the clustering can be accessed at the Eigenfactor browser: http://eigenfactor.org/projects/gender_homophily.

Given the scale of our dataset, we use Social Security and crowd sourced population records to impute (i.e., estimate the values of the missing data for the purpose of our analysis) the gender for authorships based on how these given names tend to be gendered. Due to limitations of the population-scale data, we are limited to inferring the gender of men and women authors and acknowledge the inability to include intersex and non-binary identities in our imputation. For each authorship, as in West et. al. [19], we treat gender as known if the respective first name—or one of the first names in case of double names—is used for either men only or women only at least 95% of the time. We start by using United States Social Security Administration records which allow gender imputation for 75.3% of authorships. Using genderize.io [33], which obtains gender prevalence by first name from user profiles across major social networks, we impute gender for an additional 12.6% of authorships. The remaining 12.1% of authorship instances consists of 7.6% of authorships with a name that appears in neither database and of 4.5% of authorships with a name that is used for men and women with at least 5% frequency. We provide detailed descriptive statistics in the supplement. In the main analysis, authorships with unimputed genders are omitted from our analysis, and a sensitivity analysis is given in Section 5.3.

We first consider all documents with more than one author published between 1960–2011 (560,657 documents and 1,865,835 authorships). After omitting papers which are not clustered into a terminal field and authorships which we do not impute, this amounts to 252,413 documents with 807,588 authorships.

## 4 Methods

### 4.1 Formal statistical test

To formally test the null hypothesis that the observed gender homophily can be fully explained by varying gender distributions over sub-fields and time, we compare the observed value of $\alpha$ to values generated by a *null distribution*. This distribution is constructed to encode the

structural and compositional homophily we would expect in plausible alternative configurations, but otherwise reflects the null hypothesis of no behavioral homophily. If the observed $\alpha$ is larger than most of the values generated by the null distribution, then the observed value is unlikely to occur under the modeled structural and compositional homophily. This suggests that the null hypothesis may be false and that other factors—i.e., behavioral homophily—may have caused the unusually large value of $\alpha$. Indeed, the p-value for each terminal/composite/ top-level field is the proportion of $\alpha$'s from the null distribution which are greater or equal to the observed $\alpha$. Following the standard statistical hypothesis testing framework, a small p-value (i.e., when the observed $\alpha$ is larger than many of the values from the null distribution) gives evidence against the null hypothesis.

The null distribution in Eq (1) assigns a probability to configurations of the entire JSTOR corpus where each authorship is re-assigned to a document. Roughly speaking, we fix the documents and field structure while permuting authorships so that co-authorships are formed without regard to gender. Formally, let $a$ denote an authorship in the set of all authorships **A**, let $d$ denote a document in the set of all documents **D**, and let $f$ denote a terminal field in the set of all terminal fields **F**. Let $X = \{f_a, d_a\}_{a \in A}$ be a configuration of the corpus where $f_a$ and $d_a$ denote the terminal field and document to which authorship $a$ is assigned, let $X^* = \{f_a^*, d_a^*\}_{a \in A}$ denote the originally observed configuration. Let $p_{f_a f_a^*}$ be a scalar between 0 and 1. Finally, let $y_a$ and $y_a^*$ denote the publication year of document $d_a$ and $d_a^*$ respectively, and let $Z(y_1, y_2)$ denote a function which is decreasing in $|y_1 - y_2|$. We define the gender-blind null distribution as follows where $P(X)$ is the probability of drawing some configuration $X$:

$$P(X) = \frac{\mathbb{I}_{\{X \sim X^*\}} \prod_{a \in A} [p_{f_a f_a^*} Z(y_a, y_a^*)]}{\sum_{X' \,:\, X' \sim X^*} \left( \prod_{a \in A} p_{f_a' f_a^*} Z(y_a, y_a^*) \right)}. \tag{1}$$

The equivalence relation $X \sim X^*$ indicates that $X$ is a permutation of the authorships in $X^*$ and the indicator variable $\mathbb{I}_{\{X \sim X^*\}}$ is 1 when this is true and 0 otherwise. This encodes the structural homophily in the corpus by ensuring that the null distribution only places positive probability on configurations which preserve the total authorships per terminal field, the total numbers of men and women authorships, and the number of authorships per document.

To encode compositional homophily, the null distribution does not sample permutations of the authorships uniformly, but instead draws configurations according to a probability determined by a product of terms which correspond to each individual authorship. Specifically, the term for each individual authorship is the product of two terms which govern the probability that the authorship is permuted from it's originally observed document to its newly assigned document. For any two terminal fields, $f$ and $f^*$, the quantity $p_{f, f^*}$ governs the probability that an authorship originally observed in terminal field $f^*$ will be swapped into terminal field $f$. For every pair of terminal fields, the exact value of $p_{f, f^*}$ used in our analysis is determined by observed citation flows. Details are discussed in the supplement, but at a high level, we set $0 \le p_{f, f^*} \le 1$ and $\sum_{f \in \mathbf{F}} p_{f, f^*} = 1$. Moreover, $p_{f, f^*}$ is large when $f = f^*$ or $f$ and $f^*$ are "nearby" as determined by citation flows. We set $p_{f, f^*}$ to be small or 0 when $f$ and $f^*$ are far apart as determined by citation flows. In almost all cases, $p_{f, f} > .9$, so that configurations with authorships in their original terminal fields are vastly more likely than configurations where they are far away. This ensures that the null distribution reflects compositional homophily induced by intellectual interests. However, we still allow swaps between nearby terminal-field with small probability to reflect cross-field collaborations. This also makes the null distribution less sensitive to the otherwise discrete clustering of documents to terminal fields.

To account for compositional homophily with respect to time, we also include a term, $Z(y_a, y_a^*)$ which considers the publication year of an authorship's originally observed document

and the publication year of the document it has been permuted to. By setting this term to be larger when $|y_a - y_a^\star|$ is small, we place higher probability on configurations where authorships are swapped to documents published around the same time as their originally observed document. This ensures that the null distribution reflects compositional homophily due to time. Specifically, we use:

$$Z(y_a, y_a^\star) = \begin{cases} 1 & \text{if } |y_a - y_a^\star| \leq 1 \\ \left(\dfrac{3}{4}\right)^{|y_a - y_a^\star|^2} & \text{if } |y_a - y_a^\star| > 1 \end{cases}, \tag{2}$$

so that $Z$ stays flat if the two documents differ by up to 1 year, and then decreases exponentially in the squared distance so that a probability of a swap becomes effectively 0 when $|y_a - y_a^\star| \geq 5$.

The distribution defined by (1) governs the probability of plausible alternative configurations for the entire corpus, and for each configuration, we can measure the $\alpha$ value for every terminal, composite, and top-level field. This defines a null distribution of $\alpha$ for each field, and allows us to calculate the p-value for the field. We again emphasize that when we sample values of $\alpha$ from the null distribution on a top-level/composite field, these values reflect the compositional homophily induced by the structure of the terminal fields which comprise the top-level/composite field.

We have chosen to use a finite-population approach which forms a null distribution by permuting the observed authorships. Thus, we perform inference conditioned on the observed authorships and their characteristics. In contrast, an infinite-population approach might compare the observed corpus to new resampled authorship corpuses. Using a finite-population approach allows us to avoid specifying a potentially complicated model for how the authorships and their characteristics are generated.

## 4.2 MCMC sampler

Calculating the denominator in (1) is intractable, so we sample hypothetical configurations from the null distribution indirectly using a Markov chain Monte Carlo Metropolis-Hastings sampling procedure.

We briefly describe the procedure that we use to generate draws from the null distribution; a detailed description is given in the supplement. Our sampling procedure starts with the observed configuration of authorships and generates assignments $X^{(t)}$, $t = 1, \ldots, T$ by successively modifying the current state by a series of smaller permutations. We generate a proposal for each of these cycles by first randomly selecting a length $l$. Then, authorships $\{a_1, \ldots a_l\}$ are selected to form a permutation cycle where $a_i$ is reassigned to the current location of $a_{i+1}$ and $a_l$ is reassigned the current location of $a_1$. This proposed cycle is then accepted or rejected with the appropriate Metropolis-Hastings probability.

In a sampled configuration, if a field only contains authorships of a single gender, $\alpha$ is undefined. When calculating a p-value, we consider this as $\alpha = 1$. This approach is conservative because it increases p-values, but in practice has little effect on our results.

For each sample, we start from the current assignment, $X^{(t)}$, and propose 20,000 cycles to re-assign authorships across terminal fields. This final configuration of authorships is then set as $X^{(t+1)}$, and we repeat this procedure for 150,000 samples, recording $\alpha^{(t)}$ for composite field and terminal field in our hierarchical clustering at each iteration. We repeat this procedure for three different chains, and for each chain, we discard the first 75,000 samples as burn-in. We then use the remaining 225,000 total samples (75,000 from each chain) for the main analysis.

## 5 Results

### 5.1 Main results

Table 1 summarizes results for the entire JSTOR corpus and all top-level fields. The first column gives both the observed $\alpha$ and the expected $\alpha$ from the null distribution. The expected $\alpha$ is positive for every top-level field, implying that even when collaborator choices are gender-blind, same-gender co-authorships are expected to occur more often simply because of the structure and gender composition of these fields. Also, the observed $\alpha$ exceeds the expected $\alpha$ in all top-level fields except for Mycology. Fig 3 provides a representation of the hierarchical clustering for Economics; the observed $\alpha$ is 0.11, but given only structural and composition homophily, we would expect an $\alpha$ of 0.04. Similar illustrations for all top-level, composite, and terminal fields are available in the interactive browser.

For each top-level, composite, and terminal field, we form a p-value to test the hypothesis that the observed gender homophily can be fully explained by the gender blind null distribution. This results in 1755 total p-values; however, terminal fields comprise composite fields, and in turn, composite fields comprise top-level fields. Thus in Table 1, the number of significant composite fields should not be thought of as "in addition" to the number of significant

**Table 1. Main results.** Results for the JSTOR corpus and each top-level field, sorted largest to smallest (top to bottom) by number of authorships. The $\alpha$ column gives the observed value and expected value under no behavioral homophily; the *WM* column gives the number of heterophilous documents corresponding to the observed and expected $\alpha$. The "P-value" column gives the Benjamini-Yekutieli adjusted p-value for the top-level field; "Terminal" and "Composite" columns give the counts of terminal and composite fields are which significant at the .005 level / significant at the .05 level / total.

| Field | Observed / Expected | | P-value | Sig .005 / Sig .05 / Total | |
|---|---|---|---|---|---|
| | $\alpha$ | *WM* | | **Terminal** | **Composite** |
| JSTOR | .11/.06 | 34/35.7 | 0 | 35/70/1450 | 35/50/280 |
| Mol/Cell Bio | .05/.02 | 38.2/39.4 | .00 | 8/13/178 | 11/14/44 |
| Eco/Evol | .06/.04 | 31.9/32.8 | .00 | 3/8/257 | 6/8/56 |
| Economics | .11/.04 | 18.7/20.3 | .00 | 1/5/136 | 4/5/28 |
| Sociology | .19/.10 | 38.5/42.9 | .00 | 6/10/94 | 5/9/21 |
| Prob/Stat | .09/.05 | 26/27.3 | .00 | 0/0/90 | 0/0/23 |
| Org/mkt | .16/.06 | 29/32.5 | .00 | 4/6/68 | 3/3/4 |
| Education | .16/.07 | 41.2/45.8 | .00 | 5/6/42 | 3/4/10 |
| Occ Health | .10/.03 | 41.7/44.9 | .00 | 5/8/24 | 1/1/1 |
| Anthro | .12/.07 | 38.5/40.7 | .00 | 0/1/63 | 0/0/8 |
| Law | .17/.11 | 29.7/31.7 | .01 | 0/0/98 | 0/0/16 |
| History | .16/.09 | 32.9/35.6 | .01 | 0/0/49 | 0/0/6 |
| Phys Anthro | .07/.02 | 34.7/36.3 | .00 | 1/1/32 | 0/2/10 |
| Intl Poli Sci | .09/.04 | 27.3/28.7 | .28 | 0/0/34 | 0/0/2 |
| US Poli Sci | .15/.10 | 25.2/26.7 | .04 | 0/1/37 | 0/0/6 |
| Philosophy | .10/.06 | 18.7/19.6 | .82 | 0/0/45 | 0/0/8 |
| Math | .04/.03 | 14.3/14.4 | 1.00 | 0/0/46 | 0/0/9 |
| Vet Med | .09/.03 | 38.3/40.6 | .00 | 0/5/19 | 0/0/2 |
| Cog Sci | .18/.11 | 35.7/38.7 | .00 | 1/3/14 | 1/3/3 |
| Radiation | .09/.03 | 34/36.3 | .00 | 1/3/14 | 1/1/5 |
| Demography | .15/.08 | 40.3/43.3 | .00 | 0/0/20 | 0/0/2 |
| Classics | .07/.03 | 38.7/40.6 | .84 | 0/0/35 | 0/0/8 |
| Opr Res | .03/.03 | 16.6/16.7 | 1.00 | 0/0/18 | 0/0/4 |
| Plant Phys | .08/.05 | 29.1/30.2 | .81 | 0/0/21 | 0/0/3 |
| Mycology | .03/.03 | 36.8/36.7 | 1.00 | 0/0/16 | 0/0/1 |

## Gender Homophily in Science

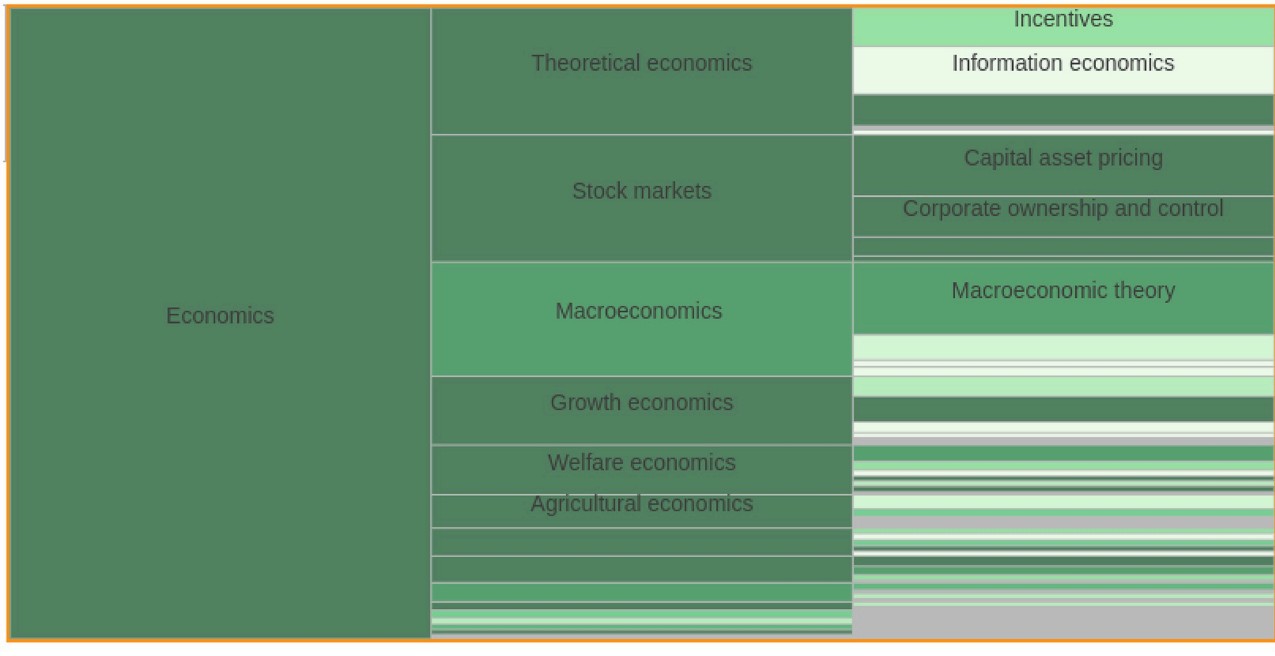

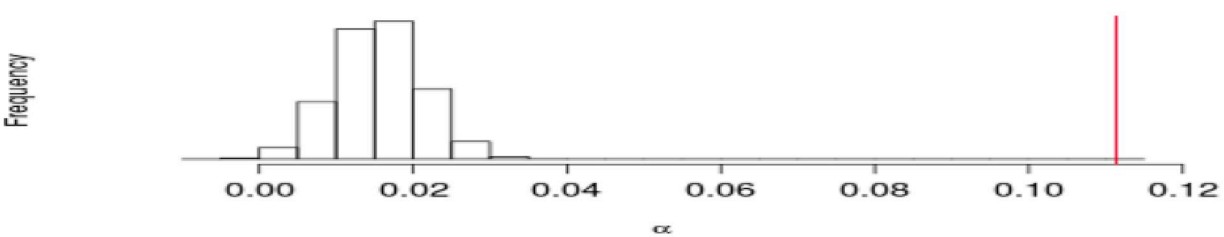

**Fig 3. Gender homophily across sub-disciplines.** The top figure shows the hierarchical clustering of economics; the height of each rectangle indicates the size relative to the top-level field and a darker shade of green indicates a smaller p-value. The bottom panel shows the histogram of $\alpha$ values when accounting for structural and compositional effects (i.e., the null distribution) with the observed $\alpha$ value indicated by the vertical red line. An interactive version of this figure for exploring other disciplines can be found in the online browser.

terminal fields (or vice versa), but rather a measurement of the same phenomena at a different level of aggregation. In addition, the p-values calculated are not independent. Thus, we adjust the calculated p-values by the Benjamini-Yekutieli [34] procedure to control the false discovery rate (FDR) at both .05 and .005. The procedure allows for arbitrary dependence across hypothesis tests, but is likely quite conservative in our setting. In the supplement, we also provide results for the less conservative Benjamini-Hochberg procedure [35]. With a .05 FDR level, we reject the null hypothesis of no behavioral homophily in the JSTOR corpus, in 17/24 (71%) top-level fields, in 50/280 (18%) of composite fields (not including the top fields), and in 70/1450 (5%) of terminal fields; the results with an FDR of .005 are also given in Table 1.

Across JSTOR, and in almost every top-level field, the incidence of significant behavioral homophily in composite fields is at least as large as the incidence among terminal fields. We posit two reasons for this. First, composite fields are an aggregation of terminal fields, and we expect that behavioral homophily in a single terminal field typically implies behavioral

homophily in the corresponding composite field. However, as seen in the Eigenfactor browser, there are composite fields with significant homophily despite having no significant terminal fields. Thus, we also posit that composite fields have higher testing power due to their larger size. This highlights the benefit of our approach which allows for valid tests in composite fields (because we account for compositional homophily) where we have more power.

At first glance, the proportion of fields where we detect significant behavioral homophily and the effect sizes we estimate (through the WM measure) may seem relatively small. For example, in Molecular and Cell Biology, the expected WM, 39.4, is 3.1% larger than the observed WM, 38.2. So— roughly speaking—we would expect 3.1% more heterophilous collaborations, when controlling for the observed gender composition of all sub-fields, if co-authorships formed randomly. However, we emphasize that our analysis explicitly controls for structural factors and conditions on the observed authorships and heterogeneous gender composition. The estimated effect sizes only reflect the incremental change which could have occurred by selecting randomly from the possible co-authors "at hand" and not the broader effect of gender homophily in co-authorship patterns. Thus, the estimated effect does not measure the gendered effects of who decides to enter and stay in academia, nor does it directly measure the effect of reasons why gender composition varies widely across scholarly fields. Also, it is true that significant behavioral homophily is detected in a relatively small proportion of terminal fields. However, as previously mentioned we would expect to have low power in terminal fields due to our non-parametric approach, and the proportion of fields where we detect significant behavioral homophily becomes substantially larger when considering composite and top level fields.

## 5.2 Secondary analysis

As a secondary analysis, we test whether certain characteristics of a terminal field are associated with significant behavioral homophily observed in the multi-author documents. Specifically, we fit a logistic regression where the outcome is whether or not significant behavioral homophily is detected ($H_i$) in a terminal field. Significant behavioral homophily is defined as having a p-value below the specified cut-off after adjustment by the Benjamini-Yekutieli FDR procedure.

Previous work [21] has shown that gender homophily is positively associated with increased women representation. Furthermore, where concerns about gender discrimination are common, we might observe a positive association across sub-fields between relative rates of solo-authorship for the lower-frequency gender (women in most cases) and increased behavioral homophily, since both would be rational choices in reaction to gender discrimination in collaboration [20, 36–38]. Thus, covariates we include are the ratio of % of solo-authorships which are women to the % authorships on multi-authored documents which are women ($R_i$), the proportion of all authorships (solo and multi) which are women ($W_i$), and the log of the number of authorships ($A_i$). Since gender dynamics may systematically differ if a field is majority woman (as opposed to the typical field which is majority man), we also include an indicator ($M_i$) for whether the field is majority woman (i.e., $M_i = 1_{\{W_i < .5\}}$) and the interaction term ($W_i \times M_i$). The full model we fit is:

$$\log \frac{Pr(H_i = 1)}{1 - Pr(H_i = 1)} = R_i + A_i + W_i + M_i + (W_i \times M_i). \tag{3}$$

Results from the logistic regression are shown in Table 2 where standard errors are calculated using a generalized estimating equation procedure with a diagonal working covariance where the clusters correspond to top-level fields. As one would expect, the estimated regression

**Table 2. Secondary analysis.** Results of the logistic regression using significant behavioral homophily with a .05 cut-off under the Benjamini-Yekutieli FDR procedure as the dependent variable. The results when using the Benjamini-Hochberg FDR procedure and .005 cut-off are included in the supplement.

| | Estimate | Robust S.E. | Robust z | P-value |
|---|---|---|---|---|
| Intercept | -15.31 | 1.63 | -9.42 | 0.00 |
| log(Authorships) | 1.45 | 0.20 | 7.11 | 0.00 |
| Proportion woman | 7.29 | 1.24 | 5.90 | 0.00 |
| Majority woman indicator | 10.37 | 5.15 | 2.01 | 0.04 |
| Ratio of solo vs multi women | 0.31 | 0.46 | 0.66 | 0.51 |
| Prop woman × Maj woman Interact | -19.00 | 9.11 | -2.09 | 0.04 |

coefficients depend on whether $H_i$ is defined by a .005 or .05 cut-off and which FDR procedure is applied. Thus, we focus on the sign of the estimates—instead of the magnitude—as they are robust to the dependent variable we select. Specifically, we see that the size of the terminal field (as measured by authorships) and the proportion of women authorships both have statistically significant positive associations with behavioral homophily; this holds regardless of cut-off or FDR procedure. The statistical significance of the indicator variable for when a field is majority woman ($M_i$) as well as the interaction term with the proportion of woman authorships ($W_i \times M_i$) vary depending on the cut-off and FDR procedure. While it is interesting to note that the estimate of the interaction term is negative, we also caution that the estimate may not be precise since the majority women indicator is only positive for less than .03 of all terminal fields.

The dependent variable in Eq (3)—whether the gender homophily in a field is statistically significant—is not directly observed or measured, but rather the outcome of a complex statistical test. Thus, the results of our regression analyses are subject to the methodological definition of significant effects that we employ. Future work will be required to tease out the robustness of our findings with respect to the methodological definition of significant homophily fields. Nonetheless, the results point towards interesting areas of future investigation which are expounded in the Discussion.

### 5.3 Sensitivity to missing gender indicators

The main analysis used gender indicators for the 87.9% of authorships with first names that are used predominantly for one gender and removed the other 12.1% of authorships (see Methods and materials). This rate of missingness compares favorably with previous studies [18], and we also further explore the impact of missingness with a sensitivity analysis using two different 10-fold multiple imputation strategies. Specifically, we see that even when imputing missing genders with a strategy to minimize gender homophily, we still detect significant behavioral homophily across the JSTOR corpus.

The first strategy (low homophily) imputes each missing indicator according to the proportions of estimated genders in its original terminal field. This assumes that there is no behavioral homophily in the missing data because the imputed genders are conditionally independent given the terminal field, providing a reasonable lower bound on the homophily we might have obtained given the full data. The second strategy (high homophily) imputes each missing gender indicator according to the proportions of estimated genders on its original document; if a document contains only unassigned authorships, we impute a single gender for all authorships according to the proportions of assigned genders for the terminal field. By construction, documents with one or less unimputed authorships are always homophilous; thus, this imputation strategy provides a reasonable upper bound on the homophily we might

**Table 3. Sensitivity analysis.** Impact of missing gender indicators. For each strategy, we show the average proportion (across the 10 imputations) of terminal, composite, and top-level fields exhibiting statistically significant behavioral homophily using the Benjamini-Yekutieli FDR control with .05.

| Analysis | Terminal | Composite | Top |
|---|---|---|---|
| Main Analysis | 0.05 | 0.18 | 0.71 |
| Sensitivity—low homophily | 0.04 | 0.15 | 0.66 |
| Sensitivity—high homophily | 0.48 | 0.79 | 1.00 |

have observed given the full data. We repeat the main analysis procedure to test for behavioral homophily in each of the imputed data sets.

As can be seen in Table 3, compared to the main analysis, the low homophily setting indeed has a lower proportion of fields with significant behavioral homophily, and the high homophily setting has a higher proportion of fields with significant behavioral homophily. However, while the results of the low homophily analysis do not drastically differ from the main analysis, the results of the high homophily analysis are substantially higher than the main analysis. We posit two potential reasons for this. First, we previously did not include authorships for which we did not estimate a gender; thus, the sensitivity analysis actually increases our "sample size" and may increase power to reject the null hypothesis. Thus, although we would expect to detect fewer fields with significant homophily in the low homophily setting, this may be partially counteracted by the increased power. Second, both strategies generally seek to impute authorship gender with no behavioral homophily. However, when a document has one or less estimated authorships, the high homophily strategy imputes genders in a directly homophilous way; this is in contrast to the low homophily strategy which does not employ an imputation strategy which is explicitly heterophilous.

For each strategy, Table 3 shows the average proportion (across 10 imputations) of fields with significant behavioral homophily. In both strategies, we assume that the observed gender proportions are good estimates of the actual gender proportions, and we do not address bias which may be induced if one gender is more likely to be unidentified than the other. Larivière [18] hand-check a sub-sample of randomly selected authorships. They find that for names for which no prior records existed, the proportions of men and women were consistent with the proportions of men and women in the classified names; however, in names which were not classified due to prevalent use for both genders, men were slightly over-represented. Additional details are discussed in the supplement.

## 6 Discussion

In this paper, we seek to test whether heterogeneity in gender composition across sub-disciplines and time can fully explain the observed gender homophily across the scholarly landscape. When controlling for the latent hierarchical structure of scholarly communities, field-specific cultures of collaboration, and changing gender representation across time, we find statistically significant behavioral gender homophily in co-authorships across the JSTOR corpus. This finding holds across all levels of granularity, from top-level scholarly fields to intellectually narrow terminal fields. Even when using the conservative Benjamini-Yekutieli false discovery procedure with a rate of .05, we detect significant behavioral homophily in 5% of terminal fields—where we would expect to have the least power—and 18% and 71% of of composite fields and top-level fields respectively—where we would expect to have more power. We also show that our findings are robust to missing gender information.

In a secondary analysis, we find that statistically significant behavioral homophily is positively associated with both field size and the proportion of women authors in the field. Scientifically, this result may seem counterintuitive on its face; however, it is not surprising from the perspective of homophily [39]: as the representation of women increases, it becomes more likely that same-gender individuals who are sufficiently compatible along other key dimensions become available as prospective co-authors. This is consistent with prior research in Economics which finds that behavioral homophily tends to be larger in sub-fields with a higher proportion of women [21]. Surprisingly, we find that the ratio of the proportion of single authored documents written by women to the proportion of women multi-authorships is not significantly associated with statistically significant behavioral homophily. Directly modeling single-author documents would require much stronger assumptions, but could provide more insight. Methodologically, the results agree with the intuition that one would expect higher testing power in fields which are larger (more authorships). In addition, we see that fields with a larger proportion of women (in practice this means closer to 50%) are also associated with a greater probability of detecting statistically significant homophily. We posit that this is due to the fact that when the gender distribution is closer to even, the set of achievable $\alpha$ values in the null distribution is generally larger as well.

Our results demonstrate that the observed behavioral gender homophily cannot be explained away by appeal to Simpson's paradox, which others have used to explain purported gender bias in graduate admissions [40] and the influence of topic choice on Black-white disparities in grant funding [41]. However, our study does not investigate causes for behavioral gender homophily. As such, it cannot adjudicate between a number of alternative explanations that remain. One possibility, informed by psychological research on stereotype threat [42], is that women choose to collaborate with other women to enhance their confidence, performance, and motivation in man-stereotyped domains [43–47]. Another possibility is that gender discrimination leads to the exclusion of women in collaborative teams with men authors and/or that, in the anticipation of such discrimination, women may preferentially seek solo authorships and behavioral gender homophilous collaborations [20, 36–38]. Because our study of behavioral homophily did not seek to unmask the beliefs and motivations driving decisions to collaborate (or not), our results are neutral with respect to them with one small exception: in contrast to the gender discrimination interpretation, our secondary analysis does not find that the ratio of the proportion of single authored documents written by women to the proportion of women multi-authorships to be significantly associated with behavioral homophily; however, further methodological development to increase the power of such an analysis is needed to firm up this result.

We believe the methodology for the main analysis represents a substantial step in understanding homophily by controlling for confounding using a hierarchical clustering to estimate latent structure. Specifically, previous work had controlled for sub-field structure—but only within a specific field [21], or had considered a broad corpus—but did not account for sub-field structure [23]. In contrast, our analysis examines gender homophily in a very broad corpus of documents spanning time and topic and accounts for sub-field structure (at a finer scale than previous work). In this article, we analyzed the JSTOR corpus, but the same methodology could also be directly applied to analyzing gender homophily in other collections (e.g., Web of Science). In addition, though we focus on gender and co-authorship, our approach is more broadly applicable to studying homophily in other contexts where confounding occurs, but can be controlled by estimating a latent structure (c.f., [48]).

Nonetheless, we recognize there are a number of other methodological issues that complicate the direct interpretation of the results and present fruitful avenues for future work.

Disambiguating authorships across documents is difficult without additional identifying information, and our analysis considers authorships rather than authors. Since individuals are more likely to co-author with previous co-authors, a future analysis with disambiguated authors could capture co-authorship dependency across documents. Unfortunately, using simple disambiguation procedures which consider author initials and surnames can drastically bias measurements of assortativity in collaborative networks [49]. In addition, incorporating metadata such as author email address may be required in order to substantially improve simple disambiguation procedures [50]. Because we consider a data set which begins in 1960, its not clear that the disambiguation procedures we could use would perform better than the baseline procedures which are shown to severely bias results. Thus, we use authorships rather than authors. This more agnostic approach has limitations, but incorporating poorly disambiguated data would also have limitations and could lead to overstatements of data quality.

In addition, because we are focused on the dynamics in collaborative work, we choose to simply exclude solo-authored documents from our main analysis. However, it is possible that a solo-authored document may be indicative of collaborative dynamics because it results from a failure to find a willing collaborator. Including solo-authored papers in our analysis would require strong modeling assumptions about the decision to write a solo-author document versus to collaborate on a multi-author document, and thus, we leave this for future work. Our analysis assumes that all authors can be placed into one of two categories: man and woman. Therefore, as noted in previous studies using tools for inferring gender (e.g., [51]), the methods are not well accommodated to intersex, transgender and/or non-binary authors. Ideally, future work will be able to better include all individuals.

The primary focus and methodological innovation of our work is accounting for compositional homophily due to confounding by sub-field membership. While we also account for compositional homophily due to publication year, co-authorship decisions are associated with many other aspects which we do not explicitly model. This includes other dimensions of social stratification including: institution, career level, race, and ethnicity. However, in order for these dimensions to induce dependence between co-authorship and gender, they must also be associated with gender (after conditioning on sub-field membership and publication year). Future work which explicitly accounts for these other social dimensions could clarify the extent to which they do or do not contribute to the observed gender homophily.

In this paper, our measure of behavioral gender homophily is neutral with respect to the intentions of authors as they choose collaborative relationships. However, given the professional advantages of collaboration, future work should explore questions related to scientists' strategic decision-making and outcomes. For example, in the short term, do women who engage in gender homophilous relationships experience higher rates of retention in the authorship pool, productivity, and impact? And, are there disciplinary differences in the benefits of employing such a strategy [52]? Future work should also explore how gender-homophilous co-authorships may shape community structure. In the long term, do gender-homophilous co-authorships give rise to gender-homophilous intellectual communities? And if so, does increasing the ratio of women in an intellectual community lead to its devaluation/impact, just as increasing the ratio of women in an occupation can decrease its prestige [53]?

Aside from the implicit or explicit consideration of gender when forming co-authorships, we take the most plausible alternative explanations for the observed gender homophily to be those invoking compositional effects due to varying gender representation across sub-fields and time. By controlling for these compositional effects, this paper demonstrates that they cannot fully explain the observed behavioral homophily we see across many fields and even in some of the smallest intellectual communities. As a result, we think that future inquiry into

potential causes for behavioral homophily would more fruitfully focus on the role of sociocultural norms and perceptions in co-authorship decisions.

## Supporting information

**S1 File. Supplemental analysis.** The $\alpha$ values and p-values for each field and all the other descriptive statistics reported in this document are openly available on our project website http://eigenfactor.org/projects/gender_homophily. Because the raw publication data are provided by JSTOR under license to the authors, requests for the raw data should be made to JSTOR directly. Code for the analysis and plots is available at https://github.com/ysamwang/genderHomophily. The supplement includes additional details on the calculations for Fig 1, the JSTOR data set, the data cleaning procedure, and the Metropolis-Hastings sampler for the null distribution. We also provide additional tables for the main analysis, secondary analysis, and sensitivity analysis, as well as additional simulation studies.
(PDF)

## Acknowledgments

We thank Jennifer Jacquet, Molly King, Shelley Correll, and Ted Bergstrom for early discussions on gender homophily. We thank Rebecca Ferrell for helpful comments on an advance copy of this manuscript.

## Author Contributions

**Conceptualization:** Y. Samuel Wang, Carole J. Lee, Jevin D. West, Carl T. Bergstrom, Elena A. Erosheva.

**Data curation:** Jevin D. West.

**Formal analysis:** Y. Samuel Wang, Carole J. Lee, Jevin D. West, Carl T. Bergstrom, Elena A. Erosheva.

**Funding acquisition:** Carole J. Lee, Jevin D. West, Elena A. Erosheva.

**Investigation:** Y. Samuel Wang, Jevin D. West, Carl T. Bergstrom, Elena A. Erosheva.

**Methodology:** Y. Samuel Wang, Carole J. Lee, Jevin D. West, Carl T. Bergstrom, Elena A. Erosheva.

**Project administration:** Y. Samuel Wang, Jevin D. West, Elena A. Erosheva.

**Resources:** Y. Samuel Wang, Jevin D. West, Elena A. Erosheva.

**Software:** Y. Samuel Wang, Jevin D. West.

**Supervision:** Carole J. Lee, Jevin D. West, Carl T. Bergstrom, Elena A. Erosheva.

**Validation:** Y. Samuel Wang, Carole J. Lee, Jevin D. West, Carl T. Bergstrom, Elena A. Erosheva.

**Visualization:** Y. Samuel Wang, Jevin D. West, Carl T. Bergstrom, Elena A. Erosheva.

**Writing – original draft:** Y. Samuel Wang, Carole J. Lee, Jevin D. West, Carl T. Bergstrom, Elena A. Erosheva.

**Writing – review & editing:** Y. Samuel Wang, Carole J. Lee, Jevin D. West, Carl T. Bergstrom, Elena A. Erosheva.

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
