## [Decision Letter · Decision Letter 0]

1 Dec 2022

PONE-D-22-16775Gender-based homophily in collaborations across a heterogeneous scholarly landscapePLOS ONE

Dear Dr. Wang,

First of all thank you for submitting your manuscript to PLOS ONE. We know it has taken us more than usual to get the required reviews we needed, but after careful consideration, we feel that it has merit but does not fully meet PLOS ONE’s publication criteria as it currently stands. Therefore, we invite you to submit a revised version of the manuscript that addresses the points raised during the review process.

We look forward to receiving your revised manuscript.

Kind regards,

Claudia Noemi González Brambila, Ph.D.

Academic Editor

PLOS ONE

Journal Requirements:

“This research was supported by Royalty Research Fund Grant #A118374 awarded to EE (PI) and CL (co-PI), NSF Grant #1735194 awarded to JW (co-PI), and NSF SMA 19-52069 to CTB.”

“This research was supported by the Univ of Washington Royalty Research Fund Grant A118374 awarded to EE (PI) and CL (co-PI) as well as NSF Grant 1735194 awarded to JW (co-PI)

https://www.washington.edu/research/or/royalty-research-fund-rrf/

https://www.nsf.gov/”

3. Please expand the acronym “NSF” (as indicated in your financial disclosure) so that it states the name of your funders in full.

“This research was supported by the Univ of Washington Royalty Research Fund Grant A118374 awarded to EE (PI) and CL (co-PI) as well as NSF Grant 1735194 awarded to JW (co-PI)”

Reviewers' comments:

Reviewer's Responses to Questions

**Comments to the Author**

1. Is the manuscript technically sound, and do the data support the conclusions?

Reviewer #1: Yes

Reviewer #2: Yes

Reviewer #3: Yes

2. Has the statistical analysis been performed appropriately and rigorously? 

Reviewer #1: Yes

Reviewer #2: Yes

Reviewer #3: Yes

3. Have the authors made all data underlying the findings in their manuscript fully available?

Reviewer #1: Yes

Reviewer #2: Yes

Reviewer #3: Yes

4. Is the manuscript presented in an intelligible fashion and written in standard English?

Reviewer #1: Yes

Reviewer #2: Yes

Reviewer #3: Yes

5. Review Comments to the Author

Reviewer #1: The authors have presented a detailed analysis of co-authorship gender homophily in JSTOR publications between 1960-2011.

They have succinctly identified the sources of these attribute taking into consideration the patterns that occurs at different levels and the compositional and structural effects that is related to the type of research, gender of the author and the research topic or field. They have also utilized a robust methodological approach in their analysis and have given sufficient reasons to address the limitations of their analysis by addressing the concerns of the previous reviewers. This manuscript would have significant contribution to scientific knowledge.

Reviewer #2: This study presents a detailed analysis of the women representation, gender homophily, in the scientific literature across a large database of documents such JSTOR. The results obtained by the authors in my opinion will be of interest for a broad community of scholars. Even though there are some fields not covered in their study (like physics, engineering and computational science), in my opinion the authors have addressed all the points raised by the previous reviewers, I recommend this manuscript for publication in PLOS ONE.

Reviewer #3: Congratulations to the authors on a very pertinent and innovative work. The paper offers a very acute empirical view on gender homophily in authorships resorting to a historically deep and disciplinary broad corpus built from the JSTOR database. The argument remains solid: gender homophily in authorships can occur due to structural features observable at different aggregation levels. The chosen method specifically allows one to understand the fine-grain nature of such a phenomenon while carefully distinguishing various sources of homophily. I find the control mechanisms and robustness checks to be very thorough and explained clearly and straightforwardly. Also, shortcomings of the data and the methodology have been clearly stated. Also, the reach and the specific meaning of otherwise ambigous terms are carefully exposed. Overall, I find it to be a very compelling piece. I only have two minor comments on the manuscript.

1.- The paper resorts to the JSTOR database and then (in lines 599 to 601) suggests that it could be employed to analyze gender homphily in other collections (such as WoS or Scopus). I am not that familiar with JSTOR database myself and this is not a very popular source for bibliometric analysis. It remains a innovative feature of this research to resort to such a data source. Still, every database has particular features that limit the reach and scope of the analysis conducted using its data. I wonder if some clarification on the reasons for choosing this database for the study would help further research. The accounts given –in lines 46 to 48; and 250 to 252— do not state any strengths and weaknesses of this data source. If this research is to be continued using input from other databases, it would be useful to know the grounds in which this data source has been selected.

2.- Some repetitions appear in the manuscript, a careful check could be useful. For instance, in page 2, it repeats "the the" (line 54) and "might might" (line 57).

6. PLOS authors have the option to publish the peer review history of their article (what does this mean?). If published, this will include your full peer review and any attached files.

Reviewer #1: No

Reviewer #2: No

Reviewer #3: **Yes**

---

## [Author Response · Author response to Decision Letter 0]

3 Jan 2023

R3.1: The paper resorts to the JSTOR database and then (in lines 599 to 601) suggests that it could be employed to analyze gender homphily in other collections (such as WoS or Scopus). I am not that familiar with JSTOR database myself and this is not a very popular source for bibliometric analysis. It remains a innovative feature of this research to resort to such a data source. Still, every database has particular features that limit the reach and scope of the analysis conducted using its data. I wonder if some clarification on the reasons for choosing this database for the study would help further research. The accounts given –in lines 46 to 48; and 250 to 252— do not state any strengths and weaknesses of this data source. If this research is to be continued using input from other databases, it would be useful to know the grounds in which this data source has been selected

Response: The JSTOR database includes a wide variety of journals across various disciplines; however, as you note, other databases may be more popular for bibliometric analysis. We choose to use JSTOR for our analysis because previous work has developed a heirarchical clustering of the JSTOR corpus based on citation networks. This yields a much finer representation of the scientific landscape as opposed to simply clustering by journal. This heirarchical clustering is crucial to our analysis. In addition, each identified cluster has been painstakingly hand labeled; this allows for interpretable results which are also useful for generating hypotheses for future work. We have added text in Section 3 to explain.

R3.2: Some repetitions appear in the manuscript, a careful check could be useful. For instance, in page 2, it repeats "the the" (line 54) and "might might" (line 57).

Response: Thank you for comment. We have carefully read through the manuscript to remove the typos.

Response: We have also included updated text for the ``funding statement'' and ``role of funder'' in the cover letter as requested. In addition, we have updated the formatting to be consistent with Plos One requirements.

---

## [Decision Letter · Decision Letter 1]

2 Mar 2023

Gender-based homophily in collaborations across a heterogeneous scholarly landscape

PONE-D-22-16775R1

Dear Dr. Wang,

We’re pleased to inform you that your manuscript has been judged scientifically suitable for publication and will be formally accepted for publication once it meets all outstanding technical requirements.

Kind regards,

Claudia Noemi González Brambila, Ph.D.

Academic Editor

PLOS ONE

Additional Editor Comments (optional):

Reviewers' comments:

Reviewer's Responses to Questions

**Comments to the Author**

1. If the authors have adequately addressed your comments raised in a previous round of review and you feel that this manuscript is now acceptable for publication, you may indicate that here to bypass the “Comments to the Author” section, enter your conflict of interest statement in the “Confidential to Editor” section, and submit your "Accept" recommendation.

Reviewer #3: All comments have been addressed

2. Is the manuscript technically sound, and do the data support the conclusions?

Reviewer #3: Yes

3. Has the statistical analysis been performed appropriately and rigorously? 

Reviewer #3: Yes

4. Have the authors made all data underlying the findings in their manuscript fully available?

Reviewer #3: Yes

5. Is the manuscript presented in an intelligible fashion and written in standard English?

Reviewer #3: Yes

6. Review Comments to the Author

Reviewer #3: This reviewer thanks the opportunity to review this great manuscript. The proposed changes are sufficient for this reviewer to recommend publication. Congratulations on such great work; this piece presents interesting findings that will be much appreciated in the field.

7. PLOS authors have the option to publish the peer review history of their article (what does this mean?). If published, this will include your full peer review and any attached files.

Reviewer #3: **Yes: **Matias F. Milia

---

## [Editor Report · Acceptance letter]

9 Mar 2023

PONE-D-22-16775R1 

Gender-based homophily in collaborations across a heterogeneous scholarly landscape 

Dear Dr. Wang:

I'm pleased to inform you that your manuscript has been deemed suitable for publication in PLOS ONE. Congratulations! Your manuscript is now with our production department. 

Kind regards, 

on behalf of

Dr. Claudia Noemi González Brambila 

Academic Editor

PLOS ONE